# Prevalence and Population Diversity of *Listeria monocytogenes* Isolated from Dairy Cattle Farms in the Cantabria Region of Spain

**DOI:** 10.3390/ani12182477

**Published:** 2022-09-19

**Authors:** Athanasia Varsaki, Sagrario Ortiz, Patricia Santorum, Pilar López, Victoria López-Alonso, Marta Hernández, David Abad, Jorge Rodríguez-Grande, Alain A. Ocampo-Sosa, Joaquín V. Martínez-Suárez

**Affiliations:** 1Centro de Investigación y Formación Agrarias (CIFA), 39600 Muriedas, Spain; 2National Institute for Agricultural and Food Research and Technology (INIA)-Spanish National Research Council (CSIC), 28040 Madrid, Spain; 3National Institute of Health Carlos III, 28222 Majadahonda, Spain; 4Instituto Tecnológico Agrario de Castilla y León (ITACyL), 47071 Valladolid, Spain; 5Servicio de Microbiología, Hospital Universitario Marqués de Valdecilla, Instituto de Investigación Valdecilla (IDIVAL), 39008 Santander, Spain

**Keywords:** *Listeria monocytogenes*, dairy cattle, manure, antimicrobial resistance, virulence, whole-genome sequencing, sequence type

## Abstract

**Simple Summary:**

The origin and prevalence of *Listeria monocytogenes* was studied in dairy cattle farms in order to examine its diversity and determine its possible persistence in manure. The utilization of manure for agricultural purposes is common in many countries. While properly treated and managed manure is an effective and safe fertilizer, foodborne illness outbreaks can occur, as many of the most prominent foodborne pathogens are carried by healthy livestock. It is, therefore, necessary to study the origin and persistence of zoonotic agents in general and of *L. monocytogenes* in particular, in order to avoid recirculation in farms and reduce risk for human populations.

**Abstract:**

*Listeria monocytogenes* is an opportunistic pathogen that is widely distributed in the environment. Here we show the prevalence and transmission of *L. monocytogenes* in dairy farms in the Cantabria region, on the northern coast of Spain. A total of 424 samples was collected from 14 dairy farms (5 organic and 9 conventional) and 211 *L. monocytogenes* isolates were recovered following conventional microbiological methods. There were no statistically significant differences in antimicrobial resistance ratios between organic and conventional farms. A clonal relationship among the isolates was assessed by pulsed field gel electrophoresis (PFGE) analysis and 64 different pulsotypes were obtained. Most isolates (89%, n = 187) were classified as PCR serogroup IVb by using a multiplex PCR assay. In this case, 45 isolates of PCR serogroup IVb were whole genome-sequenced to perform a further analysis at genomic level. In silico MLST analysis showed the presence of 12 sequence types (ST), of which ST1, ST54 and ST666 were the most common. Our data indicate that the environment of cattle farms retains a high incidence of *L. monocytogenes*, including subtypes involved in human listeriosis reports and outbreaks. This pathogen is shed in the feces and could easily colonize dairy products, as a result of fecal contamination. Effective herd and manure management are needed in order to prevent possible outbreaks.

## 1. Introduction

*Listeria monocytogenes* is a Gram-positive, non-spore-forming, motile, facultative anaerobic, rod-shaped bacterium that is an opportunistic pathogen. It causes listeriosis, a severe foodborne infection, leading to blood, brain or fetal infections in humans. Ruminants (cattle, sheep and goats), when infected, are mainly asymptomatic and excrete the bacteria in their feces, but infection can also lead to encephalitis or meningoencephalitis and abortion [1]. *L. monocytogenes* is resistant to cephalosporins, fosfomycin and fusidic acid [2]. Listeriosis is generally treated with ampicillin used alone or together with gentamicin [3]. Aminopenicillins can be substituted by cotrimoxazole, fluoroquinolones, rifampicin or linezolid. Vancomycin may occasionally be prescribed for non-meningeal infections, and erythromycin is used for listeriosis treatment during pregnancy [2]. Various recent reports have highlighted the elevation in the rate of resistance to one or more of the previously mentioned antibiotics, mainly in environmental and animal isolates and less frequently in clinical cases [4,5,6].

*L. monocytogenes* can be found in a variety of sources, including water, soil, food products, humans and animals [7,8]. At the farm level, the introduction sources of *L. monocytogenes* to livestock are not completely understood, but the route of *L. monocytogenes* infection in ruminants is considered to be contaminated silage, contaminated cattle bedding and contaminated water troughs [9]. The life cycle of this pathogen between the animals and their natural environment is fundamentally uninterrupted and hard to break [10]. As an environmental saprophyte, it is highly adapted to harsh conditions and competitive microbiota [11]. This is an important factor to take into consideration in silage production, as a pH below 6.0 should be achieved during silage fermentation in order to impede the growth of *L. monocytogenes* [12]. Many studies have reported the repeated detection of *L. monocytogenes* in samples collected from dairy farms [13,14,15]. The presence of *L. monocytogenes* in tank milk (TM) and tank milk filters (TMF) [16,17] was attributed to fecal contamination during the milking process and/or inefficient cleaning and sanitizing processes, possibly leading to biofilm formation in the milking equipment. Although on-farm eradication is highly unlikely, considering the ability of *L. monocytogenes* to survive and multiply in many habitats and hosts, the transmission and contamination rates could probably be decreased by implementing appropriate intervention strategies.

*L. monocytogenes* can be divided into four different phylogenetic lineages (I, II, III and IV), with each lineage containing different serotypes [18]. According to the somatic O antigen, it can be sub-classified in 14 serotypes. All serotypes could cause listeriosis, but serotypes 1/2a, 1/2b and 4b are more prevalent, causing over 95% of invasive infections. Serotype 4b isolates account for 50% of human listeriosis outbreaks [19] and are mostly prevalent in epidemic outbreaks. Serotypes 1/2a and 1/2b are mostly sporadic [20], with serotype 1/2a more adapted to environmental conditions [18]. Most studies suggest a higher prevalence of serotype 4b among cases of human listeriosis, but since this prevalence cannot be attributed to a higher prevalence of this serotype in food, it has been hypothesized that serotype 4b may have a higher pathogenicity compared to serotypes 1/2a and 1/2b [21].

The characterization and differentiation of foodborne pathogenic bacteria at the strain level have been achieved by the development of molecular methods. Macrorestriction with specific restriction enzymes combined with pulsed-field gel electrophoresis (PFGE) has been used since the 1990s as an effective tool for identifying bacterial subgroups. PFGE using restriction enzyme *Asc*I and *Apa*I is well established as an effective molecular subtyping method for characterizing *L. monocytogenes* [22,23]. On the other hand, the multilocus sequence typing (MLST) method for analysis of *L. monocytogenes* isolates clonality is carried out by sequencing seven housekeeping genes: *abcZ*, *bglA*, *cat*, *dapE*, *dat*, *ldh* and *lhkA* [24], and has shown that *L. monocytogenes* forms a population structured by groups of genetically similar isolates, called clonal complexes (CCs) [24]. Currently, whole-genome sequencing (WGS) is the most powerful typing tool for population biology studies, including *L. monocytogenes* [25]. Core genome multilocus sequence typing (cgMLST, covering 1748 genes in *L. monocytogenes*) is a method with a high reproducibility rate that allows inter-laboratory strain comparison by using standardized allele and type nomenclatures [26,27].

The prevalence and diversity of *L. monocytogenes* isolated from dairy farms in the Cantabria region (north coast of Spain) are described in this study. Information on the genotypic characteristics is provided and possible public health implications are being discussed, bearing in mind that information on the molecular level of the foodborne bacteria can help in anticipating outbreaks and designing effective control strategies in the event of possible disease emergence. The PCR serogroups and pulsotypes of all isolates, as well as the WGS analysis and the antimicrobial susceptibility profiles of a selection of 45 isolates of PCR serogroup IVb were performed. According to the results of this study, the *L. monocytogenes* analyzed encode virulence factors associated with human disease, capable of surviving at least 35 days in livestock waste and 70 days on pasture. The information presented here helps to increase our understanding of the on-farm life cycle of *L. monocytogenes*.

## 2. Materials and Methods

### 2.1. Study Design

The sampling was performed during 2017—2019 and included 14 dairy cattle farms (5 organic and 9 conventional) from 8 different sites in the Cantabria region (north coast of Spain). Samples were taken from forage, water, raw tank milk, the tank milk filters, fresh feces, stored manure and soil. In total, 424 samples were collected and tested for *L. monocytogenes*.

### 2.2. Sample Collection, Processing and L. monocytogenes Isolation

The samples were collected using sterile material, transported to the laboratory on ice within 2 h from the collection and subjected to detection of *L. monocytogenes* immediately upon arrival using 10 g for solid samples; 3 L filtered through a 0.5 nm sterile filter for water; 100 mL centrifuged at 8500 rpm/10 min for milk. Rectal swabs and filters from the milk tank were placed directly in the selective media. Samples were treated according to the ISO11290-1:2017 method (https://www.iso.org/standard/60313.html, accessed on 1 January 2018) with modifications. Briefly, for an initial selective enrichment, each sample was inoculated in 100 mL half Fraser broth (HFB) (Oxoid, Basingstoke, UK) for 24 h at 30 °C. A second enrichment followed, by inoculation of 100 µL of the suspension of the first enrichment in 10 mL of Full-strength Fraser broth (FFB) (Oxoid, Basingstoke, UK) for 24 h at 37 °C. Next, 100 µL were spread onto agar *Listeria* according to Ottaviani and Agosti (ALOA) (bioMérieux, Marcy-l’Étoile, France) and modified Oxford agar (MOX) (Oxoid, Basingstoke, UK) and incubated for 24 h at 37 °C. Presumptive *Listeria* spp. colonies from each ALOA and MOX plates were re-streaked to tryptic soy agar supplemented with 0.6% yeast extract (TSA-YE) plates (bioMérieux, Marcy-l’Étoile, France), incubated for 24 h at 37 °C and subjected to microscopic examination, Gram staining, glucose fermentation, oxidase, catalase and API tests (bioMérieux, Marcy-l’Étoile, France), according to the manufacturer’s instructions.

### 2.3. Decline of L. monocytogenes in Manure

The experiment was performed as previously described [28] and included a tank with manure not inoculated with *L. monocytogenes* (used as negative control), a tank with manure inoculated with *L. monocytogenes* strain CIP103575 (https://www.pasteur.fr/en/cip-distribution, accessed on 1 January 2019; positive control) and samples inoculated with *L. monocytogenes* strain MS6507 isolated from cattle feces (this study), all repeated six times (n = 6). The *L. monocytogenes* strains used were grown at 37 °C in tryptic soy broth supplemented with 0.6% yeast extract (TSB-YE) (bioMérieux, Marcy-l’Étoile, France). Before manure inoculation, *L. monocytogenes* strains were centrifuged in order to remove growth media, washed and resuspended in 0.1% peptone buffer (Oxoid, Basingstoke, UK) to a density of 10^9^ CFU/mL. Inoculants were providing 10^6^ CFU/g of manure wet weight, a quantity that represents a very high pathogen load and represents the worst-case scenario in terms of pathogen load. The presence of *L. monocytogenes* was monitored by survival curves generated by plate counting in CHROMagar^TM^
*Listeria* plates (https://www.chromagar.com/en/product/chromagar-listeria, accessed on 1 January 2019).

### 2.4. Decline of L. monocytogenes in Pasture Crops

The experiment was performed as previously described [28] using compost infected with *L. monocytogenes* MS6507. The infected compost was applied to crops and the experiment was performed in a germination chamber under controlled conditions of temperature and light [28]. The presence of *L. monocytogenes* on the surface of plants was monitored on CHROMagar^TM^ *Listeria* plates for the generation of the survival curve and the calculation of CFU/g of plant, using 3 g of pasture crops as described in [28].

### 2.5. PCR Serogroup Determination

*L. monocytogenes* isolates were initially subtyped by using a multiplex PCR serotyping to differentiate the four major *L. monocytogenes* serotypes (1/2a, 1/2b, 1/2c and 4b) into distinct PCR serogroups (IIa, IIb, IIc and IVb) [29,30].

### 2.6. Pulse-Field Gel Electrophoresis (PFGE) Typing

The clonal groups were identified by pulsed-field agarose gel electrophoresis (PFGE). The PFGE typing results were analyzed following an optimized PulseNet standardized protocol [31], as previously described [32]. *L. monocytogenes* strain H2446 was used as reference [33]. DNA extraction was carried out on agarose plugs slices in a conventional manner. DNA digestion was performed with two different restriction enzymes, *Asc*I and *Apa*I (New England BioLabs, Ibswich, MA, USA), and the generated restriction fragments were separated by electrophoresis in a clamped homogeneous electric field (CHEF-DRII) system (BIO-RAD Laboratories, Hercules, CA, USA), with the following running parameters: gradient of 6 V/cm, angle 120°, temperature of 14 °C, initial switch time 4 s, final switch time 40 s and run time 24 h. Using the Bionumerics software (Version 4.5, Applied Maths, Kortrijk, Belgium), a database of the *Apa*I and *Asc*I patterns obtained using PFGE was constructed with all the isolates of *L. monocytogenes* found in the different surveys conducted. These *Apa*I and *Asc*I patterns were used to assign a new type of combined PFGE according to the standardized optimized PulseNet protocol for *L. monocytogenes* [31] and the PulseNet Bionumerics manual (http://www.pulsenetinternational.org/protocols/bionumerics/, accessed on 1 January 2019). The Dice correlation coefficient was applied to identify similarities between the PFGE types with a tolerance of 1.5% and an optimization of 0.5%, generating a single dendrogram using the Unweighted-Pair Group Matching Algorithm (UPGMA).

### 2.7. Antimicrobial Susceptibility Test

The antimicrobial susceptibility of the 45 whole-genome sequenced isolates was determined by broth microdilution according to the Clinical and Laboratory Standards Institute (CLSI) [34]. In this procedure, a panel of 8 antimicrobial agents was used: ampicillin (AMP), ciprofloxacin (CIP), erythromycin (ERY), gentamicin (GEN), tetracycline (TET), vancomycin (VA), meropenem (MEM) and cefoxitin (FOX). All antibiotics were purchased from Sigma-Aldrich, Darmstadt, Germany. *Staphylococcus aureus* ATCC29213, *Enterococcus faecalis* ATCC29212 and *L. monocytogenes* ATCCBAA-67 were used as quality control strains. MIC breakpoints were those established by the European Committee on Antimicrobial Susceptibility Testing (EUCAST) [35]. As in the case of FOX, CIP, GEN, TET and VA, there are no interpretive criteria available for *L. monocytogenes*, we applied the breakpoint defined by EUCAST for *Staphylococcus* spp. Isolates were considered multidrug-resistant (MDR) when they showed non-susceptibility to at least one agent in three or more antimicrobial categories [36].

### 2.8. DNA Extraction and Sequencing

The total DNA from the *L. monocytogenes* isolates was purified with the DNeasy Blood and Tissue Kit (Qiagen, Hilden, Germany) and sequenced on a MiSeq device using reagents kit v3 for 2 × 300 paired-end libraries (Illumina, San Diego, CA, USA) as previously described [37].

### 2.9. Bioinformatics Analysis

The raw reads were analyzed using the pipeline TORMES^®^ version 1.0 [38]. Genome assembly was performed with SPAdes [39] and Quast [40] and genome annotation with Prokka [41]. Taxonomic confirmation was performed by using Kraken2 [42]. Additionally, 16S rRNA genes were extracted from each genome with Barrnap and used for taxonomic classification by using the RDP Classifier [43] at a confidence level of 0.8. Multilocus sequence typing was performed using an open source tool (MLST, T. Seemann, https://github.com/tseemann/mlst, accessed on 1 January 2020). Search of antibiotic resistance genes was performed by screening the genome against Resfinder [44], CARD [45] and ARG-ANNOT [46] databases by using ABRIcate (https://GitHub-tseemann/abricate: Mass screening of contigs for antimicrobial and virulence genes, accessed on 1 January 2020). Any hit with coverage and/or identity below 90% was removed. Pangenome was created with Roary [47] and FastTree [48]. The search of virulence genes was performed by screening the genome against the Virulence Factors Database (VFDB, [49]) by using ABRIcate. Any hit with coverage and/or identity below 90% was removed. Genes involved in conjugation, mobilization or genes known to be related to Pathogenicity Island LIPI-4, as well as *ami* and *aut* variants were detected using a custom blast database and using BLASTx version 2.12.0 [50]. The heatmap showing the virulence genes was made using the pheatmap package under R version 4.1.3 and RStudio RStudio 2022.02.0 (https://CRAN.R-project.org/package=pheatmap, accessed on 1 January 2020). The search of circular plasmids was performed with platon (https://github.com/oschwengers/platon). Further interrogation of integrons and conjugative plasmids was performed using Integron_Finder (https://github.com/gem-pasteur/Integron_Finder, accessed on 1 June 2022) and MacSyFinder (https://github.com/gem-pasteur/macsyfinder, accessed on 1 June 2022) (CONJscan_plasmids [51]), respectively.

### 2.10. Nucleotide Accession Numbers

The raw fastq files, annotated assembly and *L. monocytogenes* isolates information are all available under the BioProject ID PRJNA855628.

### 2.11. Statistical Analyses

All statistical analyses were carried out using GraphPad Prism version 9.4.0 (GraphPad Software, San Diego, CA, USA). To investigate the association between positive *L. monocytogenes* samples and source frequencies, the χ^2^ and Fisher´s exact test was performed. In linear regression analyses, the t-test was used to determine whether the slope of the regression line differs significantly from zero. In all cases, differences were considered statistically significant at *p* < 0.05.

## 3. Results

### 3.1. Environmental Sampling, Incidence and L. monocytogenes Isolation

A total of 424 samples were collected and processed from 14 different dairy cattle farms (5 organic and 9 conventional) in the Cantabria region (Northern Spain, Atlantic Coast). Samples were obtained from forage, silage, concentrate, water, raw tank milk and tank milk filters, fecal samples, manure and soil from within and around the stable. Fecal samples were collected from both unhealthy (diarrhea/infection/miscarriage) and apparently healthy dairy cows (without symptoms). The majority of samples (91%, n = 387) were positive for *Listeria* spp., of which 54% (n = 211) were positive for *L. monocytogenes* (Table 1). Strains were isolated by conventional microbiological plating methods and presumptive *L. monocytogenes* isolates were tested by multiplex PCR-based serogrouping, as described in Materials and Methods. Each of the 14 farms yielded at least one positive *L. monocytogenes* sample. Environmental samples obtained from sites around the farms related to feces (soil, slurry spreader and slurry drain) and fecal samples (stored manure and fresh feces) showed a higher percentage of *L. monocytogenes* and *Listeria* spp. in general in comparison with the feed samples and the raw dairy samples (χ^2^ and Fisher´s exact test *p* < 0.0001, Table 1). *L. monocytogenes* in particular, and *Listeria* spp. in general, were found in both organic and conventional farms with no statistical difference between them (χ^2^ and Fisher´s exact test *p* = 0.0960, Table 2). The full list of the *L. monocytogenes* isolates is shown in the Appendix A.

### 3.2. Transmission of L. monocytogenes

To assess the relationship of *L. monocytogenes* prevalence between sample categories, farm-level prevalence scatter plots were generated, and regression analyses were performed as described before [52]. Scatter plots were constructed for all possible relationships between samples, but only the environmental samples versus stored manure regression analysis had a slope significantly different from zero (*p* < 0.05) (Figure 1). Since the prevalence of *L. monocytogenes* in stored manure is higher than in environmental samples (Table 1) and assuming that transmission is expected to happen from high-contaminated to less-contaminated sites, these data indicate that *L. monocytogenes* is possibly transmitted through the manure to the farm ecosystem.

### 3.3. Decline in L. monocytogenes in Livestock Waste

One of the *L. monocytogenes* isolates (MS6507, isolated from rectal swab sample from cow) was selected to assess its ability to survive in manure. *L. monocytogenes* CIP103575 was used as a control strain. Liquid manure samples were inoculated, and survival curves were created. *L. monocytogenes* was detected until 28 days without enrichment. After that point, enrichment was performed before plating to confirm the presence/absence of *L. monocytogenes*, which was detected for one more week, until 35 days had passed. Negative control tanks always yielded negative results for *L. monocytogenes*. The results are shown in Figure 2A, expressed as the average ± standard deviation (n = 6).

### 3.4. Decline in L. monocytogenes in Pasture Crops

The stored manure was inoculated with *L. monocytogenes* and applied to pasture crops in a controlled laboratory setting. *L. monocytogenes* was detected until 15 days without enrichment. After that point, enrichment was performed before plating to confirm the presence/absence of *L. monocytogenes*. The results are shown in Figure 2B, expressed as the average ± standard deviation (n = 4). Maximum survival time on crops was surprisingly high, reaching 70 days.

### 3.5. PCR Serogroups

The 89% of the isolates (n = 187 out of 211) were identified as PCR serogroup IVb, 5% (n = 11 out of 211) as IIa and 6% (n = 13 out of 211) as serotype IIb. All three PCR serogroups were found in both organic and conventional farms with no statistical difference (χ*2* and Fisher´s exact test *p* = 0.2479). Farms had at least two different PCR serogroups, except one organic farm in which only IVb was found.

### 3.6. Pulsed-Field Gel Electrophoresis (PFGE)

A total of 211 *L. monocytogenes* isolates were characterized using PFGE in order to elucidate the genetic relationship among them. The PFGE profiles were analyzed and compared using BioNumerics. The PFGE analysis using *Apa*I and *Asc*I yielded 62 and 40 restriction profiles, respectively (representative PFGE profiles are shown in Figure 3A). The combination of the two restriction enzymes resulted in 64 restriction PFGE types or pulsotypes with 53% of the pulsotypes (34 out of 64) isolated from unique sites and not repeated among different farms (Figure 3B). In addition, the total of the farms showed a high diversity of pulsotypes, as more than one pulsotype was found within the same farm. On the other hand, isolates with the same pulsotype were isolated from geographically distant farms (Appendix A.

### 3.7. Genome Analysis of the L. monocytogenes Isolates and Multilocus Sequence Typing (MLST)

The molecular basis of the clonal relatedness and the virulence genes repertoire of 45 *L. monocytogenes* isolates was examined by whole genome sequencing (WGS). For the selection of the samples to be sequenced, the following criteria were employed: isolates should be of PCR serogroup IV, should represent different pulsotypes and should have been isolated from farms with information available about antibiotics use. Summary and statistics for the genome sequencing of the *L. monocytogenes* isolates are detailed in the Appendix A. The pangenome (the set of all genes that are present in the analyzed dataset) consisted of 4240 genes, with a core-genome (the pool of conserved genes, which are represented in all genomes included in the analysis) of 2545 (60%) genes; softcore-genome (the set of genes present in at least 95% of genomes analyzed) of 83 (3%); shell-genome (the pool of genes moderately common in the pangenome; more than 15% but less than 95%) of 434 (10%) and cloud-genome (the genes present in less than 15% of the genomes analyzed) of 1178 (27%). Plasmids were not detected in any of the isolates. Nevertheless, genes involved on conjugation and/or mobilization of plasmids were detected in 25 of the 45 isolates, using a custom blast database in BLASTx [50] and MacSyFinder [51]. The results are given in Appendix A. 

In silico MLST determination was performed, and showed that the 45 sequenced *L. monocytogenes* isolates belonged to one of the following 12 different sequence types (STs): ST1, ST2, ST4, ST6, ST54, ST59, ST217, ST388, ST389, ST489, ST666 and one novel sequence type ST2921 with the following housekeeping allele combination: *abcZ*(1), *bglA*(1), *cat*(12), *dapE*(659), *dat*(2), *ldh*(1), *lhkA*(3). ST1 was the most abundant (13 out of 45 isolates, followed by ST666 (7 out of 45 strains), ST54 (7 out of 45 strains), ST388 (5 out of 45 strains), ST6 (4 out of 45 strains), ST217 (3 out of 45 strains) and ST2, ST4, ST59, ST389, ST489 and ST2921 (1 out of 45 strains, each) (Table 3). In the approximately-maximum-likelihood phylogenetic trees from the alignments of the accessory and the core genes (Figure 4A,B, respectively), the isolates are grouped according to the CCs that they belong.

### 3.8. In Silico Analysis of Antimicrobial Resistance Genes and In Vitro Antimicrobial Susceptibility Testing

Only four antimicrobial resistance genes, present in all of the 45 *L. monocytogenes* isolates, were detected: *fosX* (involved in fosfomycin resistance and detected with ResFinder and CARD database), *lin* (involved in lincomycin resistance and detected with CARD and ARG-ANNOT databases), *norB* (conferring quinolone resistance and detected with CARD database) and *mprF* (which protects against cationic peptides and detected with CARD database).

The 45 isolates used for the WGS were subjected to antimicrobial susceptibility testing by using a panel of eight antimicrobial agents: ampicillin (AMP), ciprofloxacin (CIP), erythromycin (ERY), gentamicin (GEN), tetracycline (TET), vancomycin (VAN), meropenem (MEM) and cefoxitin (FOX). All isolates were susceptible to GEN, but resistant to FOX. The percentages of resistance to the rest of antimicrobial agents were as follows: AMP (15.5%, n = 7), TET (28.9%, n = 13), MEM (11%, n = 5), CIP (11%, n = 5), ERY (6.7%, n = 3) and VAN (4.4%, n = 2). Six isolates were found to be multidrug-resistant (MDR), as displayed resistance to at least 3 drugs in different antimicrobial categories (Table 4 and Appendix A.

There was no statistically significant difference in antimicrobial resistance ratios between organic and conventional farms (χ^2^ and Fisher´s exact test *p* = 0.2337). The results are given in Appendix A. Of the 6 MDR isolates, only one [MS6499(CC1)] was isolated from an organic farm; the other five [MS6484(CC1), MS6488(CC388), MS6490(CC666), MS6485(CC388) and MS6501(CC1)] were isolated from conventional farms.

### 3.9. In Silico Analysis of Virulence Genes

Virulence genes were detected in silico using TORMES^®^ version 1.0 by screening the *L. monocytogenes* isolates genomes against the Virulence Factors Data Base (VFDB) [49] and using the *L. monocytogenes* strain EGD-e as reference. For *Listeria* pathogenicity island 4 (LIPI-4) the *L. monocytogenes* strain CLIP80459 was used as reference and was detected using a custom blast database and using BLASTx version 2.12.0 [50]. The majority of virulence genes were shared across the 45 sequenced *L. monocytogenes* genomes (Figure 5). All the sequenced isolates harbored *inlA*, *inlB*, *inlC, inlJ* and *inlK*, encoding proteins of the “internalin” family of *L. monocytogenes*, which interact with distinct host receptors to promote infection of human cells [53]. An additional gene encoding a member of the internalin family, *inlF*, was also present in most isolates (39 out of 45). All sequenced isolates contained the pathogenicity island LIPI-1, a 9 kb DNA fragment composed of six genes (*prfA*, *mpl*, *plcA*, *plcB*, *actA* and *hly*), the products of which are required for the intracellular life cycle of *L. monocytogenes* [54]. Most isolates (38 out of 45) harbored a 6 kb DNA fragment corresponding to the pathogenicity cluster LIPI-3 composed of eight genes (*llsA*, *llsB*, *llsD*, *llsG*, *llsH*, *llsP*, *llsX* and *llsY*) which encodes listeriolysin O, a pore-forming toxin involved in virulence [55]. On the other hand, the pathogenicity island LIP-4, a system strongly associated with the central nervous system and placental infections, was only found in 9 isolates. Other genes involved in virulence, present in all sequenced isolates are *prsA2* gene, coding a critical post-translocation secretion chaperone [56] and loci encoding the Clp stress tolerance mediators, ClpC, ClpE and ClpP, involved in intraphagosomal survival [57,58,59].

Only two isolates [MS6670 (CC59) and MS6677 (CC489)] harbored *ami* gene, which codes for an autolysin required for adhesion to eukaryotic cells [60]. However, the rest of the isolates (43 out of 45) contained truncated versions of *ami* gene leading to a truncated protein with 772 or 607 amino acids instead of 934 as in the canonical Ami sequence (Figure 6A). The same two isolates [MS6670 (CC59) and MS6677 (CC489)] that harbored *ami* gene, contained *aut* gene also, which codes for another autolysin, required for entry into eukaryotic cells [61]. The shorter *aut* variant LMOF2365_RS00075 [62] was found in all the rest of the sequenced isolates. Four isolates belonging to CC6 (MS6492, MS6666, MS6668 and MS6669) showed a deletion of six amino acids in the shorter Aut variant (Figure 6B).

## 4. Discussion

*Listeria monocytogenes* is a facultative foodborne pathogen and several listeriosis outbreaks have been linked to its presence in the farm environment, either because of fresh vegetables fertilized with infected manure or because of the consumption of contaminated fresh dairy products [21,63]. The results of the present study demonstrate that samples from sites related to feces (slurry spreader, stable floor, slurry drain, manure and fresh feces) had a higher prevalence of *L. monocytogenes* than the ones found in feed and raw dairy samples (Table 1). These data support the hypothesis that on-farm transmission appears to be due to ingestion of feed contaminated with *L. monocytogenes* and, afterward, fecal shedding of *L. monocytogenes* in the environment by the bovine hosts, both with and without clinical disease [8]. In a way, livestock contributes to the increase and spread of *L. monocytogenes* into the farm environment, a hypothesis that is consistent with previous studies [52,64].

Several studies conducted under controlled laboratory conditions confirmed that *L. monocytogenes* can grow on plants [65,66,67]. The main drawback of most of those studies is the absence of other microorganisms. As a result, these experiments are best-case scenarios demonstrating that *L. monocytogenes* can colonize the surface of most plants. Whether or not *L. monocytogenes* colonizes plants internally is still a matter of debate, and conflicting reports are available [67,68,69]. Nevertheless, data available so far favor the hypothesis that *L. monocytogenes* can utilize nutrients from the plants to multiply and survive on plant surfaces. This hypothesis could explain the extended survival time of *L. monocytogenes* in crops, as shown in Figure 2B. Even though the experiment was performed in controlled laboratory conditions and the presence of rhizosphere and phyllosphere microbiome was limited, the maximum survival time of *L. monocytogenes* on crops was surprisingly high (70 days). The results of the decline of *L. monocytogenes* in pasture crops in combination with its decline in manure (Figure 2A) demonstrate that the life cycle of this pathogen between dairy animals and their surroundings is continuous. In order to reduce the contamination risk, effective herd management aiming the reduction of intestinal carriage, and manure treatment and management, when used as fertilizer in grassland and cropland, should be applied [8].

In this study, three PCR serogroups were isolated: IIa (lineage II) and IIb and IVb (lineage I). Comparatively few studies have isolates from natural environments and/or animals characterized to PCR serogroup and PFGE subtype. In general, according to previous studies, *L. monocytogenes* lineage I and lineage II isolates seem to be similarly prevalent [18,52,70], though contamination patterns of lineage I and II may differ [23]. Nevertheless, in this study, 95% of the isolates belonged to lineage I. Usually, linage I isolates display higher tolerance to low pH [71], which can be advantageous for the survival of *L. monocytogenes* in the gastric acidic environment and may account for their association with the fecal samples. Taking into account the prevalence of lineage I isolates in this study, it can be an additional indicator that the presence of *L. monocytogenes* in the farm environment is due to fecal contamination. Similar results were observed in poultry carcasses [23].

*L. monocytogenes* isolates were characterized using PFGE in order to elucidate their genetic resemblance. A total of 211 isolates were subtyped by PFGE, using *Apa*I and *Asc*I endonucleases resulting in 64 PFGE types or pulsotypes (Figure 3). The pulsotypes were separated by their PCR serogroup, results that are in accordance with previous studies [72,73] that confirmed relationships between serotype and PFGE patterns and reveal that the *L. monocytogenes* population of dairy farms in the Cantabria region is genetically diverse. From these 64 pulsotypes, 45 isolates were selected for whole genome sequencing, employing the following criteria: isolates should be of PCR serogroup IVb, represent different pulsotypes and be isolates from farms with information available about the antibiotics use. The reason why the PCR serogroup IVb was selected instead of IIa and IIb, which were also isolated from the dairy farms, is because the PCR serogroup IVb strains are commonly implicated in outbreaks and clinical human listeriosis cases. Previous studies showed that serotype 4b displays the highest pathogenicity compared to other serotypes in zebrafish embryos [74] and highest virulence than other serotypes according to reports that used mouse and *Galleria mellonella* as a model for identifying virulence determinants [75,76]. These features make the PCR serogroup IVb strains a very interesting candidate for further analysis at the genomic level.

A diversity of STs was found, as the 45 sequenced isolates were grouped in 12 different STs. Except for a new ST [ST2921(CC54)], isolated from an organic dairy farm, the rest of the *L. monocytogenes* isolates sequenced in this study included STs already reported. The most prevalent ST (Table 3) was ST1 (29% of all the sequenced isolates), followed by ST6 (15% of all the sequenced isolates), both of them related to outbreaks and clinical cases [77,78]. The prevalence of ST1/CC1 in ruminant-associated *L. monocytogenes* isolates has also been reported in another recent study conducted in Spain [79]. Interestingly, clones ST1/CC1, ST2/CC2, ST4/CC4, and ST6/CC6 constitute almost half of the sequenced *L. monocytogenes* isolates of this study (18 out of 45). The predominance of those clones in dairy products and ruminant feces has been already demonstrated and as they invade more efficiently the gut, they probably could lead to a higher fecal shedding and release of *L. moonocytogenes* in the environment [80]. Of particular concern is the isolation of ST388 (CC388), as it has been identified as the cause of a severe outbreak that took place in 2019 in Spain (data from the Spanish Ministry of Health). These findings highlight the importance of surveillance programs in farm animals.

Antibiotic resistance is one of the major concerns of public health since the percentage of infections due to resistant bacteria is increasing [81]. From the 45 isolates tested for their minimum inhibitory concentration (MIC), only six were characterized as multidrug-resistant. All 45 isolates were resistant to cefoxitin, an expected result, since *Listeria* spp. are naturally resistant to cephalosporins [2,82]. Overall, our results confirm the susceptibility of *L. monocytogenes*. However, some isolates were found resistant to antimicrobials used to treat human cases of listeriosis, such as ampicillin and tetracycline, and even though the antimicrobial resistance among *L. monocytogenes* isolates is still low, there are reports of increasing resistance. This is an important issue since it is known that *L. monocytogenes* can acquire and/or transfer resistance genes through horizontal transfer processes in the intestinal microbiota [83]. In a recent study, 61% of *L. monocytogenes* isolates from a meat facility contained conjugative and/or mobilizable plasmids [84]. In contrast, no plasmids were detected in the isolates described here. Nevertheless, genes responsible for plasmid conjugation and/or mobilization were detected by MacSyFinder, with protein profiles mostly belonging to the MOBP1 Hidden Markov Models (HMMs) profile (Appendix A). The MOBP1 HMMs profile targets relaxases from the MOB_P_ conjugative transfer system [85], a diverse superfamily divided in seven subfamilies (MOB_P1_-MOB_P7_) [86]. The conjugative transfer systems are classified in six MOB families: MOB_F_, MOB_H_, MOB_Q_, MOB_C_, MOB_P_ and MOB_V_, with MOB_P_ being the most diverse as it can be detected in plasmids from various incompatibility groups [86]. Since proteins from the MOB_P_ conjugative transfer system is present in many conjugative and mobilizable plasmids [87] we can only hypothesize that conjugative and/or mobilizable plasmids might be present in the isolates studied, but it was not possible to detect them with the tools used.

The presence of core genome virulence factors and pathogenicity islands was evaluated for the 45 sequenced isolates in this study (Figure 5). Four *Listeria* pathogenicity islands (LIPI) have been verified thus far in the literature, involved in the invasion, survival and colonization of *Listeria* in host tissues. Three of them have been described for *L. monocytogenes* (LIPI-1, LIPI-3 and LIPI-4), whereas LIPI-2 has only been detected in *L. ivanovii* [88]. LIPI-1 contains six genes: *hly*, *prfA*, *plcA*, *plcB*, *mpl* and *actaA* [89,90]. It encodes virulence factors that enable bacteria to escape from the vacuole (endosome or phagosome), proliferate in the cytosol and spread to the adjacent cells. LIPI-3 is composed of eight genes: *llsA*, *llsB*, *llsD*, *llsG*, *llsH*, *llsP*, and *llsX*. It encodes listeriolysin S (LLS), a haemolysin acting as a bacteriocin able to alter the host intestinal microbiota [91]. LIPI-4 encodes a cellobiose-family phosphotransfer system (PTS) and is involved in neural and placental infection [76]. Each of the 45 isolates sequenced in this study had at least one LIPI, with LIPI-1 being the one present in all strains. LIPI-3 was absent in ST2/CC2, as reported before [74], and also in CC59 and CC388. Isolates from CC4 and CC217 had all three LIPI, indicating the hypervirulence nature of those isolates. Originally the presence of LIPI-4 was reported only in strains from CC4 [76] and later other strains such as ST217/CC217 and ST388/CC388 were reported for carrying LIPI-4 [92,93], confirming the results of the present study.

In addition to the three pathogenicity islands described above, *L. monocytogenes* expresses a variety of virulence factors that are essential for its survival and persistence in the gastrointestinal tract and sequenced isolates contained various genes associated with virulence. For example, all isolates harbored the virulence genes encoding for internalins A, B, C, and J (*inlA*, *inlB*, *inlC* and *inlJ*), some cover surface proteins important in bacterial adhesion and invasion [94]. The *inlK* gene was also present in all of the sequenced isolates, encoding for a protein, InlK, which is involved in the autophagic recognition escape of *L. monocytogenes* [95]. Another gene encoding for a member of the internalin family of surface proteins, *inlF*, was found in 39 out of 45 isolates; its translational product, InlF, was reported of playing a role in *L. monocytogenes* colonization of the brain in vivo [96]. The mechanism of pathogenesis in *L. monocytogenes* involves a diversity of virulence proteins and the mechanisms leading to virulence are not completely clear yet. However, the presence of *inlA*, *inlB*, *inlC*, *inlF*, *inlJ* and *inlK* genes suggests the virulence potential of the *L. monocytogenes* isolates described in this study. Surprisingly, only two of the sequenced isolates harbored the *ami* gene, coding the Ami protein, involved in attachment to host cells and bacterial colonization of hepatocytes [60,97]. After closer inspection, all of the remaining 43 sequenced isolates not containing the *ami* gene, contained a truncated *ami* gene, leading to either a truncated 772- or a 607-amino acid Ami variants instead of the 934-amino acid wild type Ami protein (Figure 6A). Since the initial search of virulence genes was performed by screening the genome against Virulence Factors DataBase (VFDB, [49]) by using ABRIcate and accepting only hits with coverage and/or identity above 90%, the truncated version of *ami* was not originally detected but revealed after using a custom blast database and using BLASTx [50]. This deletion has been already reported by other authors [74]. The same pattern was observed for the *aut*, a gene that encodes a surface protein with autolytic activity required for invasion into eucaryotic cells and as a consequence for virulence in vitro [61]. After the first analysis, *aut* gene was found absent in all sequenced isolates, except in two isolates that also harbored the full-length *ami* gene (Figure 5). After closer inspection, it was revealed that the rest of the sequenced isolates harbored the shorter *aut* variant LMOF2365_RS00075 already reported by other authors [62]. A new variation of the *aut* allele was found in four isolates from CC6 (MS6492, MS6666, MS6668 and MS6669) showing a deletion of six amino acids (Figure 6B). Whether or not the truncated *ami* gene and *aut* allele found in the isolates described in this study lead to non-virulent or less virulent strains in vivo, remains to be tested and was beyond the objectives of the present study. In any case, our data demonstrate that dairy farms constitute a reservoir for possibly hypervirulent *L. monocytogenes*, which are shed in the feces and could easily colonize dairy products, as a result of fecal contamination.

## 5. Conclusions

An analysis of *L. monocytogenes* isolated from environmental sites of livestock and livestock farms has shown that several *L. monocytogenes*, responsible for human infection, circulate in the biosphere and agricultural systems and might contribute to the spread of these pathogens throughout the food chain, thereby posing a major health challenge. A significant portion of isolates recovered are from the same CCs as those frequently isolated from human clinical cases and outbreaks on a global scale and many of those strains encode virulence factors associated with serious illness. As the SARS-CoV2 outbreak has demonstrated, the spread of infectious diseases in humans from animal reservoirs represents a major public health risk and it is expected that zoonotic diseases will occur more often due to climate change. This is the main pillar of the “One Health” approach, which accepts that human health is tightly connected with animal health and the environment. Considering the ability of *L. monocytogenes* to survive and adapt in different ecosystems and hosts, which makes it an important archetype of the “One Health” axis, effective herd and manure management and respect of the animal welfare are needed to prevent foodborne illnesses. Farmers should pay attention especially to the transmission from animal to animal through fecal-oral routes, usually via manure contamination of the pasture or silage with the microorganism. Feeding the animals with good quality silage, avoiding any other rotten vegetation and isolation of sick animals following good hygiene and sanitation on the farm is also important. Thereafter, food safety programs throughout the food production chain (from farm to fork) are needed to prevent foodborne illnesses.

## Figures and Tables

**Figure 1 animals-12-02477-f001:**
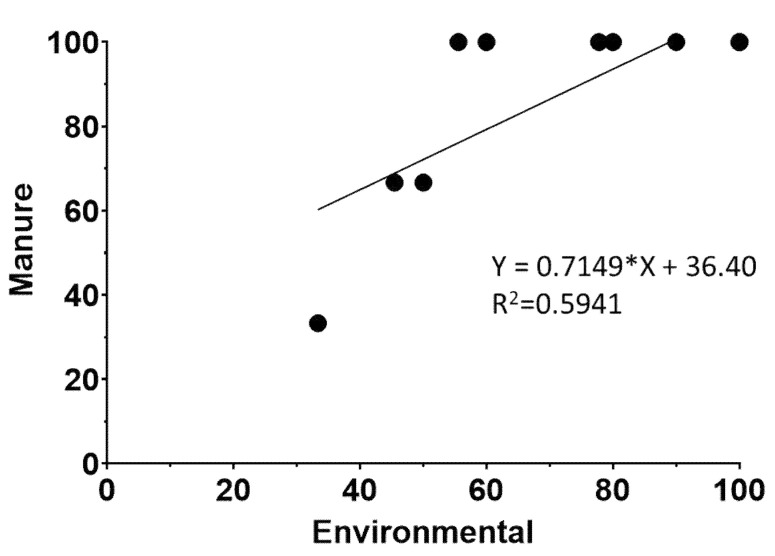
Scatter plot of percentage of *L. monocytogenes* positive manure samples versus *L. monocytogenes* positive environmental samples. Regression equation and R^2^ values are determined by regression analysis as described in Material and Methods. The (*) in the equation corresponds to the multiplication sign.

**Figure 2 animals-12-02477-f002:**
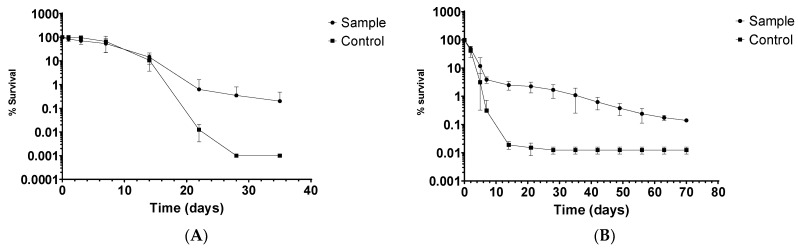
Death curve of *L. monocytogenes* in (**A**) livestock waste and (**B**) crops fertilized with infected manure.

**Figure 3 animals-12-02477-f003:**
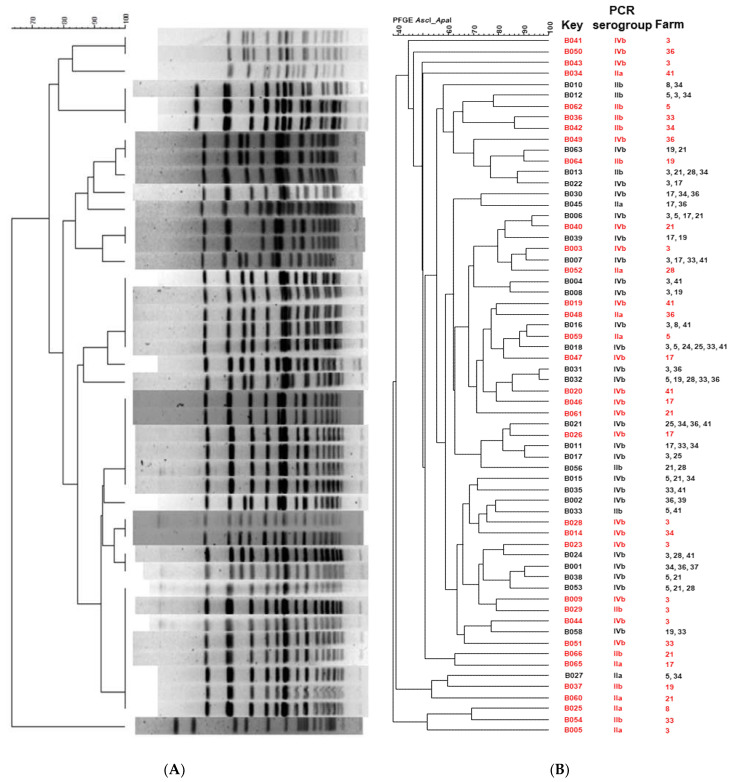
(**A**) Representative *L. monocytogenes* PFGE profiles obtained with *Apa*I. (**B**) Dendrogram showing similarities among the total number of the *L. monocytogenes* PFGE profiles obtained with *Apa*I and *Asc*I. In red are shown pulsotypes found in unique sites (not repeated between farms).

**Figure 4 animals-12-02477-f004:**
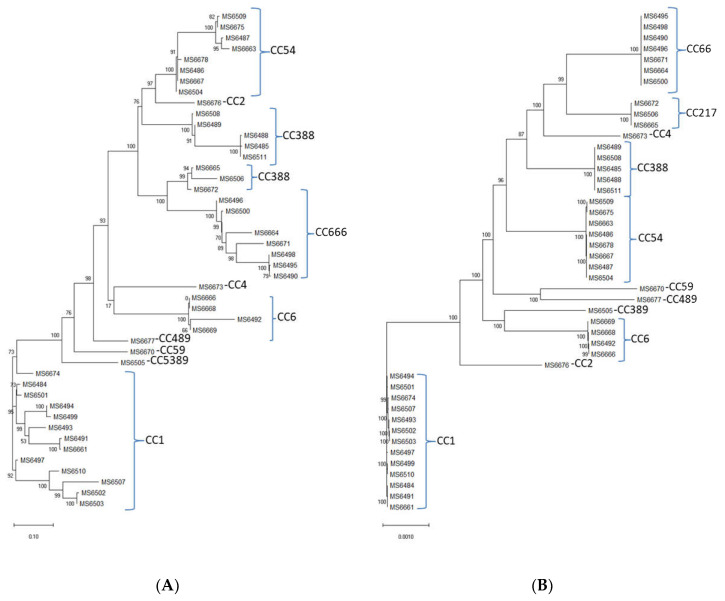
The approximately-maximum-likelihood phylogenetic trees from the alignments of the (**A**) accessory and the (**B**) core genes of the *L. monocytogenes* isolates.

**Figure 5 animals-12-02477-f005:**
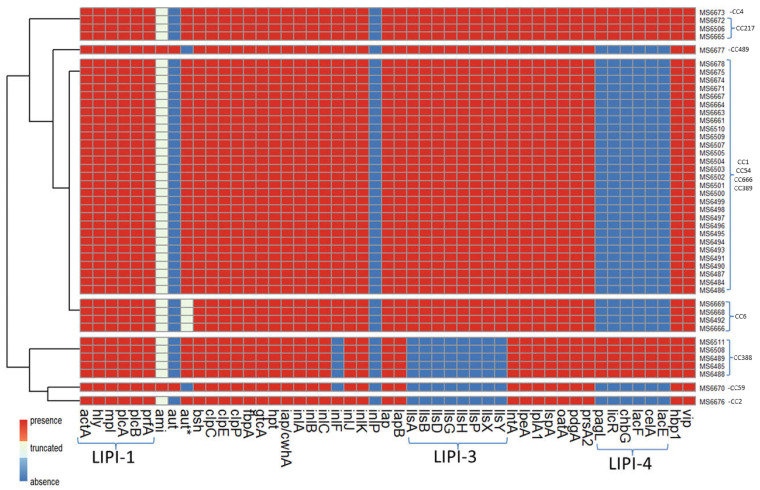
Presence/absence of virulence genes of *L. monocytogenes* isolates. *aut** refers to the short variant of *aut*; LIPI-1: *Listeria* pathogenicity island 1; LIPI-3: *Listeria* pathogenicity island 3; LIPI-4: *Listeria* pathogenicity island 4.

**Figure 6 animals-12-02477-f006:**
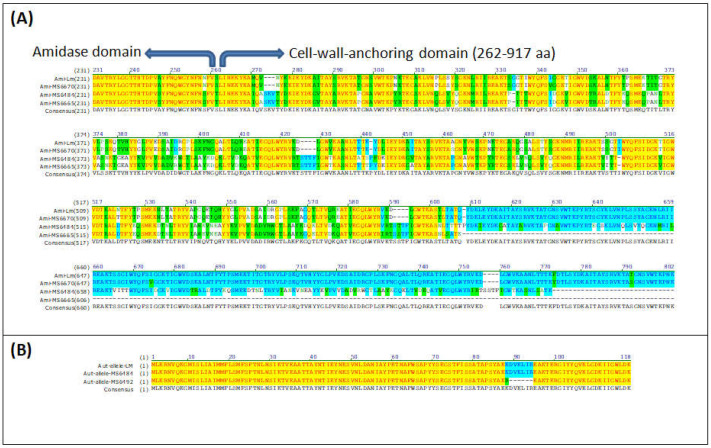
Sequence alignment of (**A**) Ami and (**B**) Aut with the respective variants found in the *L. monocytogenes* isolates. The figure shows a CLUSTALW alignment. Colour code: red on yellow background=identical residues; Black on green background = similar residues; Green on white background=weakly conserved residues; Blue on light blue background: strongly conserved residues; Black on white background = non-conserved residues. Accession Numbers are: Ami = NP_466081.1; Aut (short variant) = WP_003734189.1.

**Table 1 animals-12-02477-t001:** Prevalence of *Listeria* isolates from different samples from dairy cattle farms in the Cantabria region.

Sample Type	No of Samples Tested	No (%) of Samples Negative for *Listeria*	No (%) of Samples Positive for *Listeria*	No (%) of Samples Positive for *L. monocytogenes*
Slurry tanker	19	1 (5%)	18 (95%)	13 (68%)
Feeding through/corridor	36	1 (3%)	35 (97%)	29 (80%)
Stable floor	40	1 (3%)	39 (97%)	32 (80%)
Slurry drain	19	0 (0%)	19 (100%)	16 (84%)
**Total environmental samples**	**114**	**3 (3%)**	**111 (97%)**	**90 (79%)**
Dry forage	42	3 (7%)	39 (93%)	10 (24%)
Fresh grass crops	24	3 (12%)	21 (88%)	11 (46%)
Maize cured forage	11	0 (0%)	11 (100%)	3 (27%)
Grass cured forage	8	0 (0%)	8 (100%)	0 (0%)
Concentrate for dairy cows	41	2 (5%)	39 (95%)	12 (29%)
**Total feed samples**	**126**	**8 (6%)**	**118 (94%)**	**36 (29%)**
Filter from the milk tank	28	9 (32%)	19 (68%)	4 (14%)
Raw milk from the tank	34	10 (29%)	24 (71%)	2 (6%)
**Total raw dairy samples**	**62**	**19 (31%)**	**43 (69%)**	**6 (10%)**
Chicken manure	9	1 (11%)	8 (89%)	7 (78%)
Stored dairy manure	39	0 (0%)	39 (100%)	36 (92%)
Fresh feces from unhealthy dairy cows (diarrhea/infection/miscarriage)	33	3 (9%)	30 (91%)	28 (93%)
Fresh feces from healthy dairy cows (without symptoms)	41	3 (7%)	38 (93%)	8 (19%)
**Total fecal samples**	**122**	**7 (6%)**	**115 (94%)**	**79 (65%)**
**Total samples**	**424**	**37 (9%)**	**387 (91%)**	**211 (50%)**

**Table 2 animals-12-02477-t002:** Prevalence of *Listeria* isolates from organic and conventional dairy cattle farms in the Cantabria region.

Farm Type	No of Samples Tested	No (%) of Samples Negative for *Listeria*	No (%) of Samples Positive for *Listeria*	No (%) of Samples Positive for *L. monocytogenes*
Organic	147	9 (6%)	138 (94%)	83 (56%)
Conventional	277	28 (10%)	249 (90%)	128 (46%)
**Total samples**	**424**	**37 (8%)**	**387 (91%)**	**211 (50%)**

**Table 3 animals-12-02477-t003:** In silico multilocus sequence typing (MLST) of the 45 sequenced *L. monocytogenes* isolates. In bold, isolates from organic farms.

Isolates	ST	CC	Lineage
MS6484, **MS6491**, MS6493, **MS6494**, MS6497, **MS6499**, MS6501, **MS6502**, MS6503, **MS6507**, MS6510, MS6661, MS6674	1	1	I
MS6676	2	2	I
MS6673	4	4	I
MS6492, MS6666, MS6668, **MS6669**	6	6	I
MS6486, MS6487, MS6504, **MS6509**, MS6667, **MS6675**, MS6678	54	54	I
MS6670	59	59	I
**MS6506**, MS6665, **MS6672**	217	217	I
MS6485, MS6488, MS6489, **MS6508**, MS6511	388	388	I
MS6505	389	389	I
MS6677	489	489	I
MS6490, MS6495, MS6496, MS6498, **MS6500**, MS6664, **MS6671**	666	666	I
**MS6663**	2921 (novel)	54	I

**Table 4 animals-12-02477-t004:** Antimicrobial susceptibility of *L. monocytogenes* isolates characterized as multi-drug resistant (MDR). Values correspond to minimum inhibitory concentrations (MIC) and expressed in mg/mL.

Isolate	AMP	CIP	ERY	GEN	TET	VAN	MEM	FOX
MS6484	2^(R)^	4^(R)^	2^(R)^	0.5^(S)^	4^(R)^	4^(R)^	1^(R)^	16^(R)^
MS6488	1^(S)^	2^(R)^	2^(R)^	0.5^(S)^	2^(S)^	1^(S)^	0.25^(S)^	32^(R)^
MS6490	2^(R)^	0.5^(S)^	1^(S)^	0.5^(S)^	4^(R)^	4^(R)^	0.25^(S)^	64^(R)^
MS6485	0.25^(S)^	2^(R)^	1^(S)^	0.5^(S)^	4^(R)^	1^(S)^	0.25^(S)^	32^(R)^
MS6499	2^(R)^	2^(R)^	1^(S)^	0.25^(S)^	4^(R)^	1^(S)^	0.25^(S)^	64^(R)^
MS6501	4^(R)^	0.5^(S)^	0.25^(S)^	0.5^(S)^	2^(S)^	1^(S)^	0.5^(R)^	16^(R)^

AMP: Ampiccillin, CIP: Ciprofloxacin, ERY: Erithromycin, GEN: Gentamicin, VAN: Vamcomycin. MEM: Meropenem, FOX: Cefoxitin; (R): Resistant, (S): Susceptible.

## Data Availability

Genome sequences information is available in http://www.ncbi.nlm.nih.gov/bioproject/, (accessed on 1 June 2022) under the BioProject ID PRJNA855628.

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
