# Peer review of "Prevalence and Population Diversity of Listeria monocytogenes Isolated from Dairy Cattle Farms in the Cantabria Region of Spain"

_animals, 2022, doi:10.3390/ani12182477_

Round 1

Reviewer 1 Report

Animals-1858490

This manuscript analyzed prevalence and genetic and molecular diversity of Listeria monocytogenes isolates from dairy cattle farms in Spain. This manuscript is very well-organized and presented well with detailed corresponding tables and figures. Although this study didn’t show any new virulent strain with many virulence factors or extensive antimicrobial resistance, it will give better understanding of the bacteria. The information from this manuscript will be educational for not only the dairy industry, will be also helpful to the “One Health” approach.

Title: Prevalence and population diversity of Listeria monocytogenes isolated from dairy cattle farms in the Cantabria region of Spain. -> Please delete the ‘.’

Abstract: Please add a sentence for the conclusion. (from the conclusion part, or like the one in the simple summary)

Line 120: Please indicate the period of collecting samples. People will want to compare their findings with yours.

Line 128: 3 1 filtered – Please check the numbers and revise it.

Line 133-140: Please check again the symbols(ºC)  used in the manuscript. (->℃)

Line 153, 166: CHROMagarTM -> CHROMagarTM (superscript)

Line 201: VA -> VA, (add ,)

Line 290, 297, 315, 343: as described in Materials and Methods -> Please delete this phrase.

Line 290-291: survival curves were performed. -> survival curves were created.

Line 325: Analysis of PFGE types was performed as described in Materials and Methods. -> Please delete this sentence.

Line 341: as detailed in Materials and Methods -> Please delete this phrase.

Line 343: Please combine two sentences into one sentence. (For example, performed as describe in Materials and Methods. The MLST analysis showed -> performed and it showed)

Table 3: It is hard to tell which isolates are in which ST or CC and how many isolates are in the same group. Please add a line or numbers of isolates in the same group.

Table 4: If you used MIC values in the table, please indicate that (MIC) in the title of the table and there should be a unit (mg/ml) for them.

Line 388, 406: Please add a full name of LIPI at its first appearance in the manuscript and in the figure.

Line 407: variant of aut -> variant of aut.

Discussion: If you didn’t find any significant differences between isolates from organic and conventional farms, what about others? If you can find any similar studies, please add a paragraph about bacterial isolates from the ‘organic’ environment. Since you made groups (organic and conventional) in the abstract, many readers will read the paper expecting discussion part regarding that.

Line 450: high -> Please add the maximum survival time (day #) you observed.

Line 501: Please add (a) reference(s).

Line 512: Please re-write the sentence. (For example, there were no plasmids detected …)

Line 514: Please explain MOBP1

Line 582: if you are explaining the spread of disease from animal to human, the word ‘emergency’ can be replaced by ‘outbreak’ or other words.

Author Response

Response to Reviewer 1

First of all, we would like to thank you for your comments and for your time for revising our manuscript,

Please find below our response to your comments

Title: Prevalence and population diversity of Listeria monocytogenes isolated from dairy cattle farms in the Cantabria region of Spain. -> Please delete the ‘.’

Full stop (.) deleted

Abstract: Please add a sentence for the conclusion. (from the conclusion part, or like the one in the simple summary)

A sentence has been added at the end of the abstract (lines 47-48)

Line 120: Please indicate the period of collecting samples. People will want to compare their findings with yours.

The period of sampling has been added (line 126)

Line 128: 3 1 filtered – Please check the numbers and revise it.

Numbers have been checked and they are correct (3 l, stands for 3 litres)

Line 133-140: Please check again the symbols(ºC)  used in the manuscript. (->℃)

Thank you very much for your pointing this out. Symbols have been corrected

Line 153, 166: CHROMagarTM -> CHROMagarTM (superscript)

Corrected

Line 201: VA -> VA, (add ,)

The comma has been added

Line 290, 297, 315, 343: as described in Materials and Methods -> Please delete this phrase.

Phrase deleted

Line 290-291: survival curves were performed. -> survival curves were created.

“Performed” has been changed to “created”

Line 325: Analysis of PFGE types was performed as described in Materials and Methods. -> Please delete this sentence.

Phrase deleted

Line 341: as detailed in Materials and Methods -> Please delete this phrase.

Phrase deleted

Line 343: Please combine two sentences into one sentence. (For example, performed as describe in Materials and Methods. The MLST analysis showed -> performed and it showed)

The two sentences have been combined

Table 3: It is hard to tell which isolates are in which ST or CC and how many isolates are in the same group. Please add a line or numbers of isolates in the same group.

Lines added to the table

Table 4: If you used MIC values in the table, please indicate that (MIC) in the title of the table and there should be a unit (mg/ml) for them.

MIC has been indicated in the table header and the units have been added

Line 388, 406: Please add a full name of LIPI at its first appearance in the manuscript and in the figure.

Full name of LIPI added

Line 407: variant of aut -> variant of aut.

Full stop (.) has been added

Discussion: If you didn’t find any significant differences between isolates from organic and conventional farms, what about others? If you can find any similar studies, please add a paragraph about bacterial isolates from the ‘organic’ environment. Since you made groups (organic and conventional) in the abstract, many readers will read the paper expecting discussion part regarding that.

Thank you very much for your comment.

When we were preparing the manuscript we looked for similar studies with Listeria monocytogenes but we were not able to find any. There are many studies about Listeria monocytogenes strains isolated from dairy farms, but they do not compare antibiotic resistance or prevalence between organic and conventional farms. That is why we didn´t comment anything about that in the Discussion. After your comment, we searched again but once more, without any results. We believe that we don´t have enough information from other studies in order to discuss that part, so we would prefer to keep discussion as it is.

Line 450: high -> Please add the maximum survival time (day #) you observed.

Maximum survival time added

Line 501: Please add (a) reference(s).

Reference added

Line 512: Please re-write the sentence. (For example, there were no plasmids detected …)

Sentence changed

Line 514: Please explain MOBP1

An explanation has been added (lines 523-529)

Line 582: if you are explaining the spread of disease from animal to human, the word ‘emergency’ can be replaced by ‘outbreak’ or other words.

“Emergency” has been replaced by “outbreak”

Reviewer 2 Report

This is a very complete and well presented study on the presence and survival of L. monocytogenes in the dairy farm environment. It offers valuable information on the role of farm animals as reservoirs of listeriosis.

Accept as is.

Author Response

Thank you very much for your comments and your time revising our manuscript.